# A novel approach towards a histone replacement system in Tetrapods

**Maximilian Pfisterer** (ID)[1]*, **Amelie Pritz**[1], **Anna Parry** (ID)[1], **M. Lienhard Schmitz** (ID)[1,2]*

**1** Institute of Biochemistry, Justus-Liebig-University, Giessen, Germany, **2** Member of the CPI and the German Center for Lung Research, Giessen, Germany

* maximilian.pfisterer@biochemie.med.uni-giessen.de (MP); lienhard.schmitz@biochemie.med.uni-giessen.de (MLS)

## Abstract

Histone replacement systems are valuable tools for studying histone modifications, but in vertebrates this is so far only possible by labor-intensive CRISPR base editing of each single histone gene. To facilitate such studies, we developed an alternative method and conducted proof-of-principle experiments using histone 2B (H2B) as an example. This method relies on shRNA targeted degradation of endogenous histone mRNA and simultaneous re-expression of a replacement histone. Due to their limited histone gene number, chicken cells proved suitable as a tetrapod model system for this approach. In the first-generation system we developed an shRNA design tool that identified shRNAs leading to efficient downregulation of each of the canonical histones using a single doxycycline (Dox)-inducible shRNA. As H2B knockdown cells are not viable, they were rescued by simultaneous Dox-inducible re-expression of H2B. Since the comparability between wild-type and mutated histone variants is limited due to random chromosomal integration, we developed a second-generation "inducible knockdown-re-expression" system. This new version employs genomic landing pads to facilitate Bxb1 recombinase-mediated cassette exchange with mutated histones. The system successfully rescued reconstituted cells from histone depletion-induced cell death, but requires optimization to align histone expression levels with endogenous levels. Overall, this system offers a promising new method for studying specific histone modifications in chicken cells.

## Introduction

Eukaryotic DNA is packaged into chromatin through its association with histone octamers composed of two copies each of H2A, H2B, H3, and H4 [1]. The linker histone H1 binds to the DNA between nucleosomes, known as linker DNA, thereby stabilizing the nucleosome and facilitating higher-order chromatin compaction [2,3]. Histones regulate all chromatin-dependent processes such as transcription, replication, and DNA repair via post-translational modifications (PTMs), which either directly

**Data availability statement:** All relevant data are within the paper and its Supporting Information files.

**Funding:** Our work is supported by funding from the German Research Foundation GRK 2573/2 (RP10, Project 416910386), the Excellence Cluster CPI (Project 390649896), the EU (COST Action CA23119 Senescence2030) and the Forschungscampus Mittelhessen (2025_1_01). The funders had no role in study design, data collection and analysis, decision to publish, or preparation of the manuscript.

**Competing interests:** The authors have declared that no competing interests exist.

affect chromatin structure or act as molecular signals for other factors such as transcription factors and chromatin remodelers. These modifications form an epigenetic "histone code" which is written, erased and interpreted by chromatin-associated proteins [4]. While the existence of the histone code is widely accepted and many modifications are clearly associated with specific functions, the functional relevance of numerous detected histone modifications remains unclear. This is partly due to the inherent complexity of the histone code, as histone modifications can influence one another, function cooperatively, and exhibit context-dependent effects based on their chromatin environment [5]. Compounding this challenge is the fact that current methods used to investigate these relationships are subject to technical limitations [6]. Research on histone modifications typically focuses on correlations between the presence of specific PTMs and phenotypic outcomes, such as chromatin condensation or transcriptional activity. This approach has yielded a solid understanding of several key modifications, such as the association of the histone marks H3 lysine 27 trimethylation (H3K27me3) and H3K9me3 with heterochromatin, as well as H3K36me3 and H3K4me3 with transcriptional activation [6–8]. On the other hand, many other histone modifications remain poorly understood, this is especially relevant for modifications not directly involved in transcription, such as numerous mitotic histone PTMs [9], including H2B serine 6 phosphorylation [10,11].

To more directly investigate the function of such histone PTMs, many groups have developed histone replacement systems that seek to replace the endogenous wild-type histone of a cell with a mutant version. While this replacement approach is often applied to study the function of amino acid residues of different proteins, histones have so far been difficult to replace. The first histone replacement systems were established for unicellular eukaryotes, namely the budding yeast *Saccharomyces cerevisiae* [12–15], the fission yeast *Schizosaccharomyces pombe* [16] and for the protozoan *Tetrahymena thermophila* [17,18]. In these organisms, histones are encoded by only a few genes per histone, making genomic editing techniques easily applicable. Subsequent studies also enabled the replacement of histones in the multicellular organisms such as *Drosophila melanogaster* [19–21]. This was feasible due to the repetitive nature of histone genes in *Drosophila*, with around 100 repeats of the five canonical histones forming the locus. A histone cluster replacement system was developed by creating a histone null mutant, removing the endogenous histone expression locus [20]. Modified histone repeat units were then integrated to achieve full replacement. In the plant *Arabidopsis thaliana*, a histone replacement system developed by Corcoran et al. (2022) used an approach based on copy number reduction. A CRISPR/Cas9-mediated knockout approach reduced the number of histone H4 copies from 8 to just 1, which the plants surprisingly tolerate despite impaired growth [22]. In this experimental system, the final copy is then knocked out after integrating a replacement histone. These histone replacement systems have undergone further improvements over the years and have provided valuable insights into the functions of histone modifications [23].

In vertebrates, replacement of histone variants is already possible, as they are typically encoded by only one or two genes [24–28]. A straightforward histone

replacement system for core histones in vertebrates, however, is not available, as the various clusters encoding canonical histone genes are dispersed over multiple genomic regions and therefore cannot be easily replaced. Furthermore, the DNA sequences encoding histones can be quite divergent, thereby precluding the targeting of all histones with a single shRNA. In the only study published to date investigating the function of core histone modifications, Sankar and colleagues mutated all 28 alleles of the mouse H3 genes to examine the role of H3K27 modification [29].

While this elegant study demonstrates the feasibility of such experiments in mice by *in situ* mutation of endogenous histones, we aimed to pursue a complementary and simpler approach by developing a comprehensive tetrapod histone replacement system with the following key features:

- Applicability to all canonical histones

- Targeting of all amino acid residues

- Inducible switch from endogenous to replacement histones

- Compatibility with combinations of simultaneous histone replacements

- Rapid generation of modified cells to enable high-throughput screening approaches

## Results

### The first-generation tetrapod histone replacement system

**General considerations.** To establish a partial histone replacement system, we chose a strategy based on coupling shRNA-mediated knockdown with the re-expression of shRNA-resistant histone variants. Similar experiments have previously been successfully applied to multi-copy genes such as Ubiquitin [30]. Such a knockdown and re-expression system for histones would consist of two main components, as schematically shown in Fig 1A:

(I)  An shRNA construct that specifically targets and silences the endogenous histone to be replaced.

(II)  A replacement histone that is resistant to the shRNA.

These two components can be delivered on two plasmids, both of which are stably integrated into the genome of the target cells and allow inducible downregulation of the endogenous histone and re-expression at the same time (Fig 1B). We opted for an inducible expression system, as it enables the expression of potentially deleterious histone mutants under controlled conditions. Upon induction, expression of the shRNA targeting the endogenous histone results in a gradual reduction of the corresponding mRNA and the synthesized histone protein. Simultaneous induction of the shRNA-resistant replacement histone compensates for this loss (Fig 1C). The replacement histone can either be the wild-type version or a mutant of interest. Induction of the system leads to predominant expression of the replacement histone, while the existing endogenous histone protein will be progressively diluted by the newly synthesized replacement histone. The total histone content is normally doubling with each cell cycle during the S-phase [31]. Consequently, a theoretical maximum of 50% of histones can be replaced per cell cycle. After a few replication cycles, the majority of original wild-type histone would be largely substituted by the mutant version, depending on the overall replacement efficiency (Fig 1D). Epitope tagging of the replacement histone allows for discrimination between the exogenous replacement histone and the endogenous protein, thereby enabling further downstream experiments such as Western blot analysis or immunoprecipitation (Fig 1E).

For inducible expression, we employed a Dox-dependent system characterized by its tight regulation, enabling both efficient shRNA-mediated knockdown and upregulation of the replacement histone [32,33]. A histone-targeting shRNA should ideally target all (or at least most) mRNAs encoded by the different gene copies of a specific histone. RNAi mediated knockdown of the endogenous histone partially relies on perfect base-pairing between the effector siRNA and the

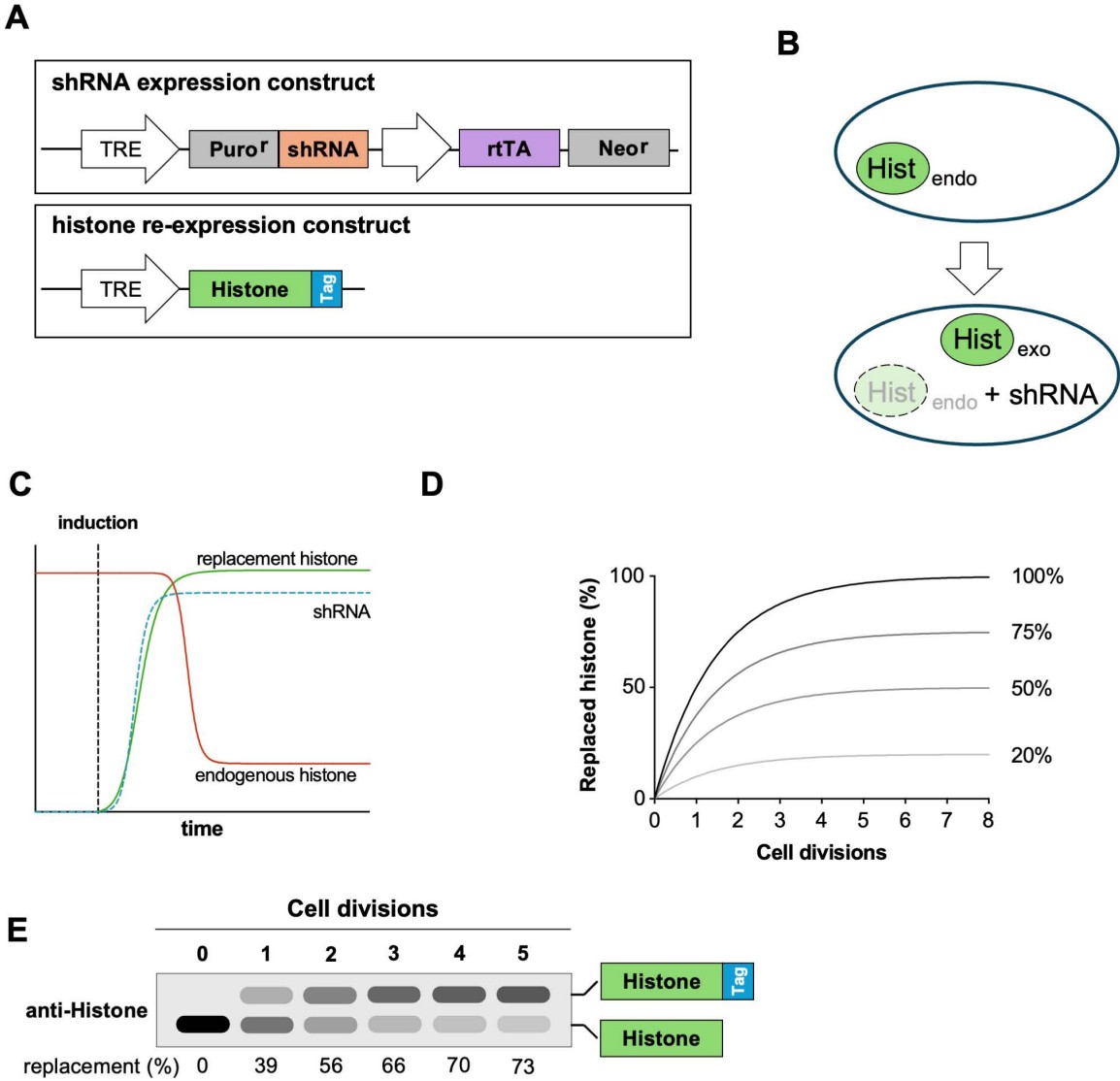

**Fig 1. First-generation histone replacement strategy. (A)** Schematic representation of an inducible histone replacement system consisting of one plasmid expressing a histone-targeting shRNA and a second plasmid expressing an shRNA-resistant replacement histone fused to an epitope tag. **(B)** First, a parental cell line expressing the shRNA is generated, leading to downregulation of the endogenous histone (Hist $_{endo}$). Subsequently, re-expression of the exogenous shRNA-resistant histone (Hist $_{exo}$, in the wild-type or mutant form) occurs, largely replacing the endogenous histone. **(C)** Induction of the system with Dox triggers both the expression of the shRNA and the mRNA expression of the replacement histone. The predicted time-dependent changes in the expression levels of endogenous and re-expressed histone proteins are shown. **(D)** With each replication, the amount of histone in the cell is doubled. At a replacement efficiency of 100%, at most 50% of the total histone can be replaced per replication. This leads to gradual dilution of the endogenous histone over time. At lower replacement efficiencies, the maximum amount of histone that can be replaced is reduced accordingly. **(E)** Schematic Western blot illustrating replacement over time with an assumed replacement efficiency of 75%. The upper band corresponds to the tagged replacement histone.

target mRNA [34]. The large number of core histone genes in the human genome is accompanied by significant divergence in the coding DNA, which prevents efficient targeting of all endogenous histone gene products by a single shRNA. Remarkably, compared to humans, chickens have significantly fewer histone genes and additionally a much higher sequence similarity within the histone-coding DNA regions (Table 1).

**Table 1. Comparative analysis of the human and chicken genomes with respect to the number of histone genes and the similarity of their isoforms.**

| Histone | Human | | Chicken | |
|---------|---------|--------------------|---------|--------------------|
| | # genes | isoform similarity | # genes | isoform similarity |
| H1 | 7 | 37% | 6 | 82% |
| H2A | 16 | 60% | 9 | 93% |
| H2B | 22 | 75% | 8 | 90% |
| H3 | 14 | 65% | 8 | 76% |
| H4 | 15 | 58% | 8 | 99% |

Since chickens represent a suitable model system for the analysis of genetic and epigenetic regulation [35], we first conducted proof-of-principle experiments in chicken cells to establish whether the naturally high expression of histones can be reduced using shRNAs. We performed these experiments using H2B as an example for a core histone, which in chicken is encoded by only 8 genes displaying approximately 90% sequence identity [36].

### Designing the H2B-targeting shRNA plasmids and the re-expression plasmid

Initially, shRNAs targeting the chicken H2B mRNAs were designed according to the sequence requirements described in [37] and further shRNA design tools (BLOCK-iT™ RNAi Designer 2021; InvivoGen siRNA Wizard 2021). Based on these criteria, a code was developed that enables rapid identification and ranking of candidate shRNAs for H1, H2A, H3 und H4. A total of seven shRNAs were selected to test their ability to downregulate H2B (see S1 Table). As shRNA #3 was identified as the most effective shRNA (data not shown), it was used in all subsequent experiments. To test the efficiency of different shRNA expression backbones, the commonly used human microRNA mir30a shRNA sequence [38] was compared to a shortened synthetic mir30 sequence [39] as well as the chicken ortholog of mir30a. Since test experiments showed no difference in knockdown efficiency between the different backbones (data not shown), the shortened mir30 sequence was chosen for further use due to its smaller size. A schematic structure of the transfected plasmid allowing Dox-inducible downregulation (pHREP-KD-1, Histone replacement knockdown) is shown in Fig 2A.

This plasmid allows two levels of selection: (I) The Neomycin resistance gene allows for G418-based selection of cell clones with the stably integrated plasmid. (II) Because cells might evade induction of shRNA expression or its processing – as it presents a selective disadvantage – an additional selection marker was introduced which allows for the quick eradication of cells not expressing the shRNA. For this purpose, the Puromycin (Puro) resistance gene was chosen to be co-expressed with the H2B-targeting shRNA.

Since histones are highly expressed proteins in proliferating cells, the corresponding shRNAs presumably also have to be produced in large amounts. To achieve this goal and to simplify stable integration, the sleeping beauty (SB) transposon system was employed [40]. Therefore, the shRNA expression construct contains inverted terminal repeats to enable integration at multiple genomic loci via the SB transposase [41,42]. In addition, compatible self-destructing restriction sites were added upstream and downstream of the miRNA backbone, allowing for the insertion of additional shRNAs (e.g., for knockdown of multiple histones) or for multimerization, if required [43], as shown in Fig 2B.

The second plasmid pHREP-RE-1 (Histone replacement re-expression of the first generation) allows for expression of the shRNA-resistant H2B replacement histone and is schematically shown in Fig 2C. The histone is linked to a His-tag and a tandem OLLAS-tag [44], which enables experimental distinction from endogenous histone through gel electrophoresis and allows for specific detection of the re-expressed H2B. Generally, tagging of H2B, even with larger tags such as a full GFP, is well tolerated *in vivo* [45]. The OLLAS-tag was chosen because it contains fewer charged amino

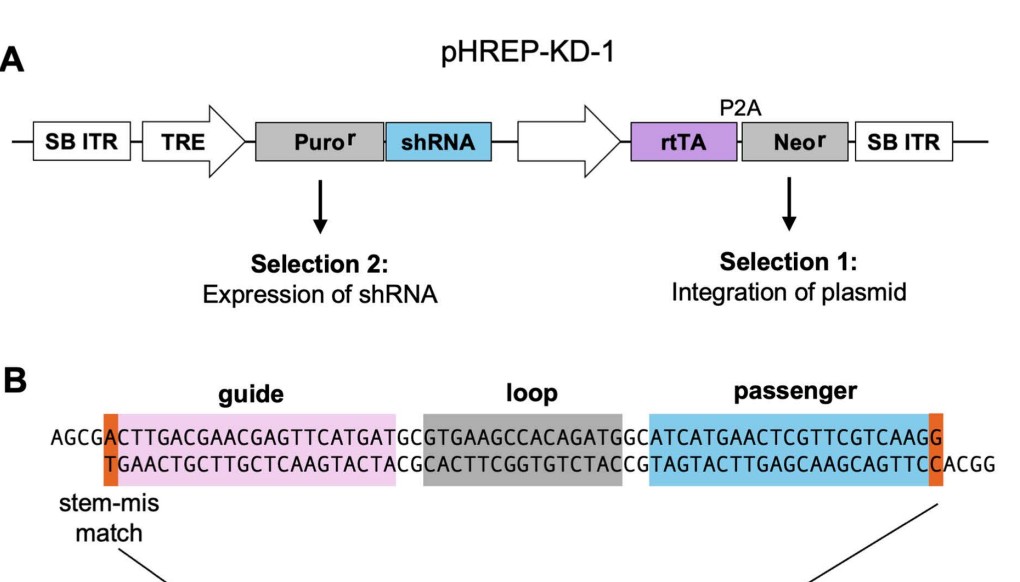

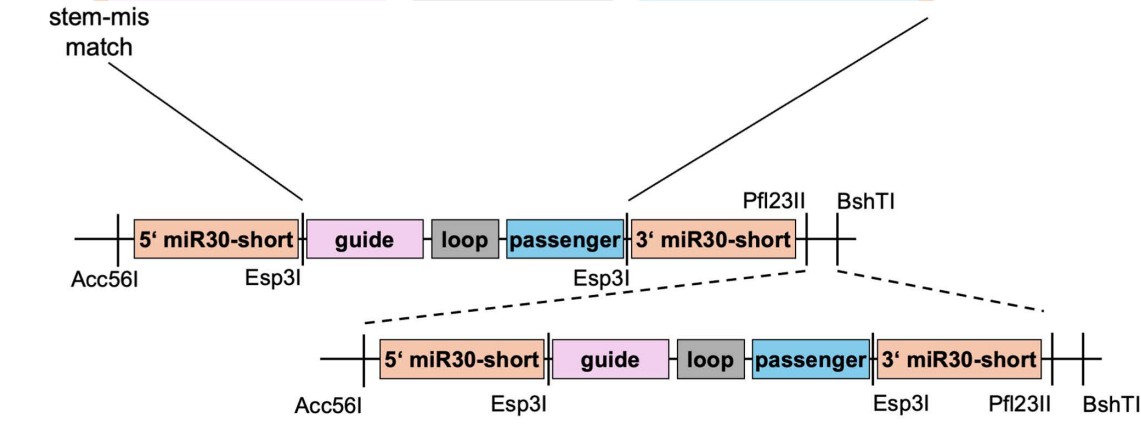

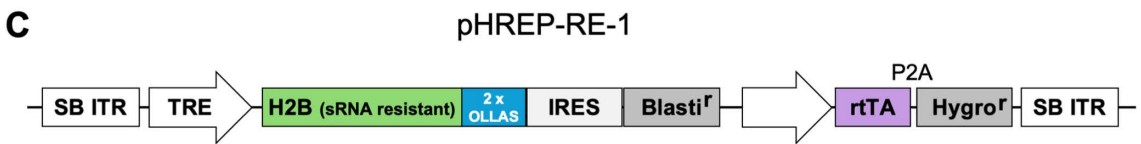

**Fig 2. Design of the plasmids for the first-generation histone replacement system. (A)** Schematic representation of the first-generation shRNA-expressing plasmid pHREP-KD-1. The plasmid contains a tetracycline-responsive promoter (TRE) driving the expression of a puromycin resistance gene and the shRNA. It also includes a constitutively expressed tet-activator (rtTA-VP16) to enable expression from the TRE. The plasmid is flanked by Sleeping Beauty inverted terminal repeat sequences (SB ITR) to allow for transposase-mediated genomic integration. Puro$^r$ = puromycin resistance, Neo$^r$ = neomycin resistance. **(B)** Schematic representation of the shRNA expression backbone using a shortened mir30a backbone, as described [39]. Esp3I restriction sites allow insertion of shRNAs via Golden Gate cloning. An example sequence targeting H2B is shown. The shRNAs can be multimerized using compatible self-destructing restriction sites. **(C)** Schematic representation of the first-generation histone re-expression construct pHREPR-RE-1. A TRE drives expression of an shRNA-resistant histone tagged with a (His)$_6$ tag and a tandem OLLAS tag to distinguish between endogenous and replaced histones. An additional expression cassette for rtTA enables induction of the TRE. The construct is flanked by SB ITR sequences to allow for transposase-mediated integration. Blasti$^r$ = blasticidin resistance, Hygro$^r$ = hygromycin resistance.

acids compared to other frequently used tags such as Flag or V5. The Dox-inducible polycistronic transcript encodes the replacement H2B, followed by an IRES sequence and a Blasticidin resistance gene to allow for the selection of expressing cells. Like the shRNA expression construct, the histone re-expression construct contains SB inverted terminal repeats (ITRs) to facilitate integration of multiple copies into the host genome to meet the histone demand of the cell.

## Generation of chicken cell lines allowing Dox-inducible downregulation of H2B

To generate chicken cells that allow for shRNA-mediated knockdown of H2B, three selection steps were applied sequentially:

(I)  G418 resistance: This allows for the selection of cells with integrated shRNA plasmids.

(II) Puromycin resistance: The addition of Dox induces the expression of a shRNA together with a Puromycin resistance gene. This selects for cells that are capable of Dox-inducible shRNA expression.

(III) Cell proliferation: Strong knockdown of a specific histone impairs DNA packaging and compromises proliferation.

This strategy is schematically illustrated in Fig 3A and was implemented using DF-1 chicken fibroblasts. These cells were transfected with the shRNA plasmid and a plasmid expressing the SB transposase for stable integration. One day after transfection, the cells were treated with G418 to select for those with the plasmid stably integrated into the genome. After 14 days of selection, 96 single-cell clones were isolated. Populations derived from these clones were then treated with Puro and/or Dox for 4 days and cell layers were stained with crystal violet. Two clones were selected that did not proliferate in the presence of Dox, whereas their growth was normal in the absence of Dox (Fig 3B).

## Generation of chicken cell lines allowing Dox-inducible downregulation and re-expression of H2B

To generate H2B replacement cell clones, one of the shRNA expression clones was transfected with a plasmid for the inducible expression of an shRNA-resistant H2B, along with a plasmid expressing the SB transposase. As with the shRNA expression plasmid, this allows stable integration at multiple genomic loci, mimicking the organization of endogenous histone genes [46]. These cells were selected with hygromycin for 14 days to obtain a stable cell pool, from which single-cell clones were subsequently isolated and expanded. The clones were examined for their survival after induction of the shRNA and the replacement histone. Cells were cultured with or without Dox, selected with Puro, and survival after 4 days was measured using a crystal violet staining, as shown in Fig 4A. While most clones showed poor survival, four re-expression clones continued to proliferate and were further tested for histone replacement.

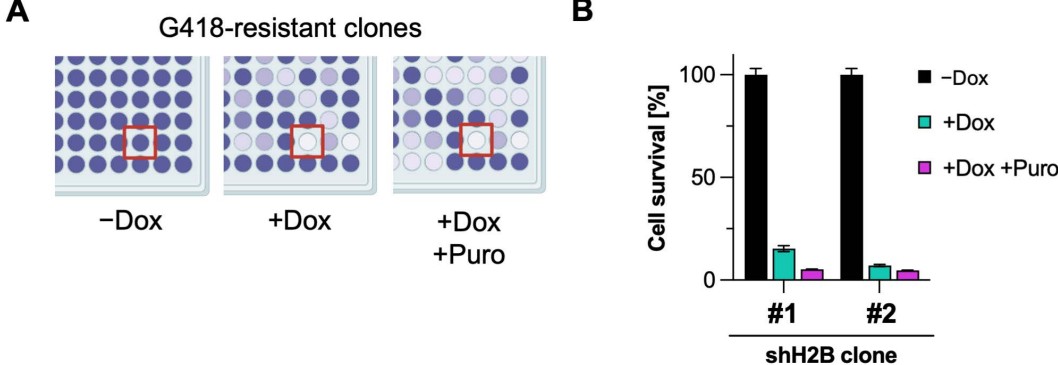

**Fig 3. Generation and analysis of stable inducible chicken H2B knockdown clones in the first-generation system. (A)** DF-1 chicken fibroblasts were transfected with the first-generation chicken H2B-targeting shRNA plasmid (pHREP-KD-1-shH2B) together with a Sleeping Beauty expressing plasmid and selected for 2 weeks with G418. Resistant single-cell clones were grown and split into 4 different 96-well plates. One plate was used to maintain the cells for later use, while the other three plates were treated with Dox and/or Puro (each at 1 μg/mL for 4 days). Cell survival was determined by crystal violet staining, a schematic result is shown. The red box marks a clone that grows in the absence of Dox but stops growing after induction of the histone knockdown, making it a good candidate for efficient knockdown. **(B)** After the initial selection shown in **(A)**, two selected shRNA-expressing clones were treated with Dox (1 μg/mL, 4 days), additionally selected with puromycin, or left untreated. Cell survival was scored by quantitative crystal-violet staining. Data from three independent biological replicates were normalized to the untreated controls and are presented as means ± SD.

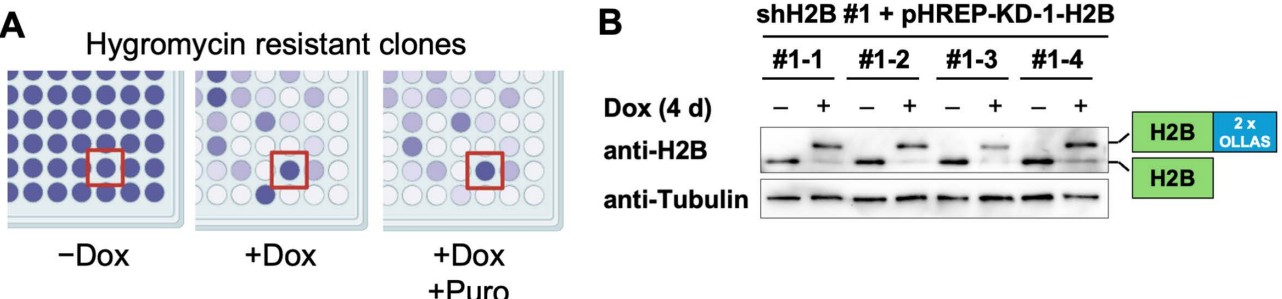

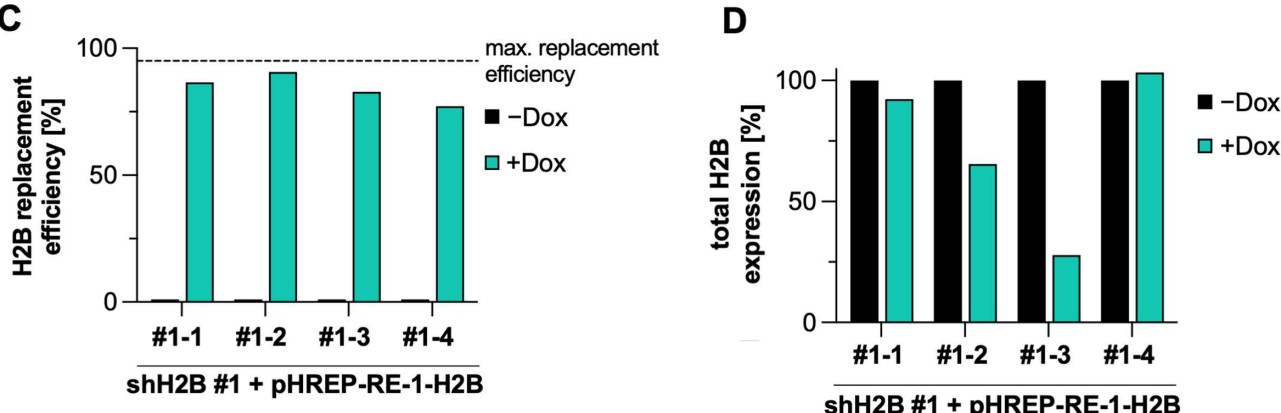

**Fig 4. Characterization of stable H2B re-expression cell clones in the first-generation system. (A)** The single clones characterized in (3B) were transfected with the first-generation wild-type H2B replacement plasmid (pHREP-RE-1-H2B) together with a Sleeping Beauty expressing plasmid. After selection with hygromycin for 14 days, a cell pool was generated and single cells were expanded. The cells were then split into different 96-well plates, as shown in Fig. 3A. The cells remained untreated or were treated with Dox +/- Puro (each at 1 µg/mL for 4 days). Determination of by cell numbers using crystal violet staining is schematically shown. **(B)** The indicated 4 selected stable replacement cell clones were treated with Dox (1 µg/mL) for 4 days as specified. Cell lysates were analyzed for the expression of endogenous and replacement H2B by Western blot, a representative result is shown. **(C)** The replacement efficiency of the blot shown in (B) was quantified as the ratio of the upper (replacement) histone H2B to the total H2B amount (replacement + endogenous H2B) and plotted. **(D)** The total H2B expression level from the blot shown in (B) was quantified and normalized to the untreated control.

To determine the efficiency of histone replacement, the four cell clones were induced again for 4 days, followed by the analysis of H2B expression by Western blot. Endogenous H2B had almost completely disappeared after induction, and in all clones a second band corresponding to the tagged replacement H2B appeared (Fig 4B). Quantification of the ratio of replacement H2B to total H2B showed that the replacement rate exceeded 75% in all clones, with clone #1–2 reaching nearly the theoretical maximum of 95% after 4 days of induction (Fig. 4C). In two of the four clones, the total H2B level after Dox induction was comparable to the total histone level under non-inducible conditions (Fig 4D) showing that the re-expression system meets the cell's histone demand. To investigate whether the other canonical histones (H1, H2A, H3, and H4) can also be efficiently downregulated, additional shRNAs were designed by a self-written program according to the criteria described [37]. The eight best ranked shRNAs were then cloned into shRNA expression plasmids and co-expressed with the respective EGFP-tagged chicken histones in 293T cells. Analysis of histone expression by RT-qPCR showed a very high knockdown efficiency of most of the tested shRNAs (S1 Fig).

In summary, it can be concluded that all chicken histones can be efficiently downregulated using a single specific shRNA, and a re-expression approach, as described here for H2B, is technically feasible. However, a few points need optimization in order to develop a reliable tool for scientific applications.

### The second-generation tetrapod histone replacement system

**Optimizing the plasmids for histone-targeting shRNAs and histone re-expression.** To optimize the vector driving the expression of inducible shRNAs and replacement histone, we utilized an improved tet-activator (rtTA-V16) with higher sensitivity and stronger transactivation capacity [47]. This amended version for Dox-inducible downregulation of the second generation (pHREP-KD-2) is displayed in Fig 5A. First, the optimal shRNAs identified here for targeting all

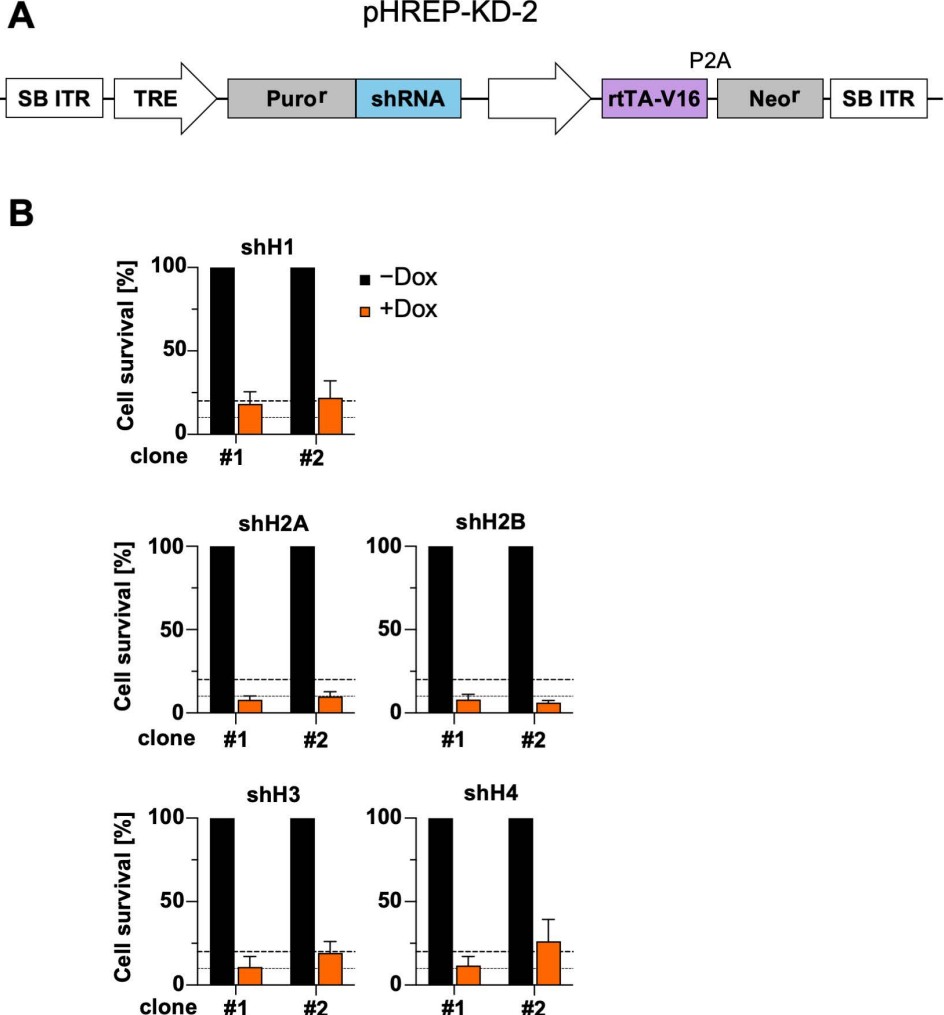

**Fig 5. Design and functional testing of the second-generation knockdown plasmids. (A)** The pHREP-KD-2 plasmid has a similar structure to the first-generation plasmid, with the difference of an optimized Tet transactivator. **(B)** The procedure follows the protocol described in Fig. 3A. DF-1 chicken fibroblasts were transfected with pHREP-KD-2 and a Sleeping Beauty-expressing plasmid. G418-resistant clones were initially functionally screened by measuring cell survival after shRNA induction with doxycycline (1 µg/mL, 4 days) compared to uninduced cells. Clones showing the greatest reduction in cell survival after induction were selected. Survival data from 3 independent biological experiments are shown as mean±SD for 2 shRNA-expressing clones per shRNA. Dashed lines indicate 10% and 20% cell survival.

canonical histones were cloned into the new shRNA plasmid. Subsequently, DF-1 chicken fibroblasts were transfected with the corresponding plasmids together with a plasmid expressing the SB transposase to enable genomic integration of the shRNA. The cells were selected with G418 for two weeks. Single-cell clones were expanded and selected based on their remaining proliferation after the addition of Dox and Puro, as described for Fig. 3. This yielded two cell clones for each of the canonical histones, which died within four days after induction of the histone-specific shRNA (Fig 5B).

Integration of payloads by transposases such as SB occur at random, with a high likelihood of integrating into active genes. By integrating replacement-histone mutants at random positions, comparability between the generated cell clones is limited, as schematically displayed in Fig 6A. Therefore, we designed a second-generation histone replacement plasmid using a recombinase-based approach to generate comparable cell clones that are isogenic except for the expressed histone. As this plasmid contains so-called landing pads (LPs) with two Bxb1 attP recombination sites [48], it was designated pHREP-LP. After random integration of the plasmid at various genomic sites, Bxb1-mediated recombination with a donor plasmid containing the complementary attB sites [49] allows for insertion of the desired wild-type or mutant histones via recombinase mediated cassette exchange (RMCE) [50], as schematically illustrated in Fig 6B. Through this strategy, both the wild-type histone and the mutant histone are integrated at exactly the same genomic sites, enabling full comparability of functional experiments. Transposase-mediated integration often results in multiple LPs per cell, therefore RMCE should lead to exchange in all rather than only a few LPs (Fig 6C). To allow sorting of cells where complete cassette exchange in all locations has taken place, the plasmid also includes a constitutive promoter that drives the expression of a polycistronic transcript encoding different selection markers. These include the surface marker LNGFR (a truncated low-affinity nerve growth factor receptor), suitable for magnetic cell separation [51] and a fusion protein of EGFP and the hygromycin resistance gene (Fig 6D). Cells with the stably integrated plasmid can be easily identified due to their Hygromycin resistance and EGFP expression, which allows FACS sorting. Additionally negative sorting of cells which still express LNGFR from an untargeted LP can be applied. BpiI restriction upstream of the last Bxb1 recombination site allow for the integration of histone re-expression cassettes in various copy numbers using Golden Gate assembly. This is used to add between 1 and 4 copies of the Dox-inducible replacement histone expression cassette (Fig 6E) to allow for fine tuning of histone expression levels and to select for clones in which the LPs have been integrated in sufficient number and at favorable locations. The replacement histone was tagged with a splitGFP tag instead of the previously used OLLAS tag, as splitGFP can be used for chromatin visualization in live-cell microscopy through complementation with the larger GFP fragment [52,53]. Cells with the stably integrated pHREP-LP plasmid expressing a wild-type histone can then be used for RMCE upon delivery of the Bxb1recombinase together with an exchange plasmid encoding the mutant histone. As schematically shown in Fig 6E, the resulting cells which have successfully undergone recombination and express the mutant histone can be identified by the gain of Zeocin resistance and the lost expression of LNGFR and the fusion protein between EGFP and the hygromycin resistance gene.

## Histone replacement using the second-generation histone replacement system

As with the first generation, reconstitution with exogenous histone was exemplarily tested with H2B. First, plasmids containing one to four H2B wild-type expression cassettes, respectively, were constructed (Fig 7A). DF-1 chicken fibroblasts with an efficient knockdown of H2B (clone #2) were transfected with these various LP plasmids containing different H2B copy numbers along with a plasmid expressing the SB transposase. Stable cell pools were generated by two weeks of selection with hygromycin. EGFP-expressing cells were subsequently sorted by FACS to obtain stable cell pools. Analogous to the characterization method chosen in the first-generation histone replacement system, cell survival of the reconstituted cell pools was measured after the addition of Dox for induction of the replacement system. Quantitative assessment of cell survival showed increased survival with each additional H2B copy per plasmid, reaching nearly 60% with the plasmid containing 3 copies of H2B. Interestingly, survival was lower in cell pools with the 4-copy plasmid (Fig 7B). Since the 3x-H2B pool showed the highest overall survival rate after induction, single cell clones were generated

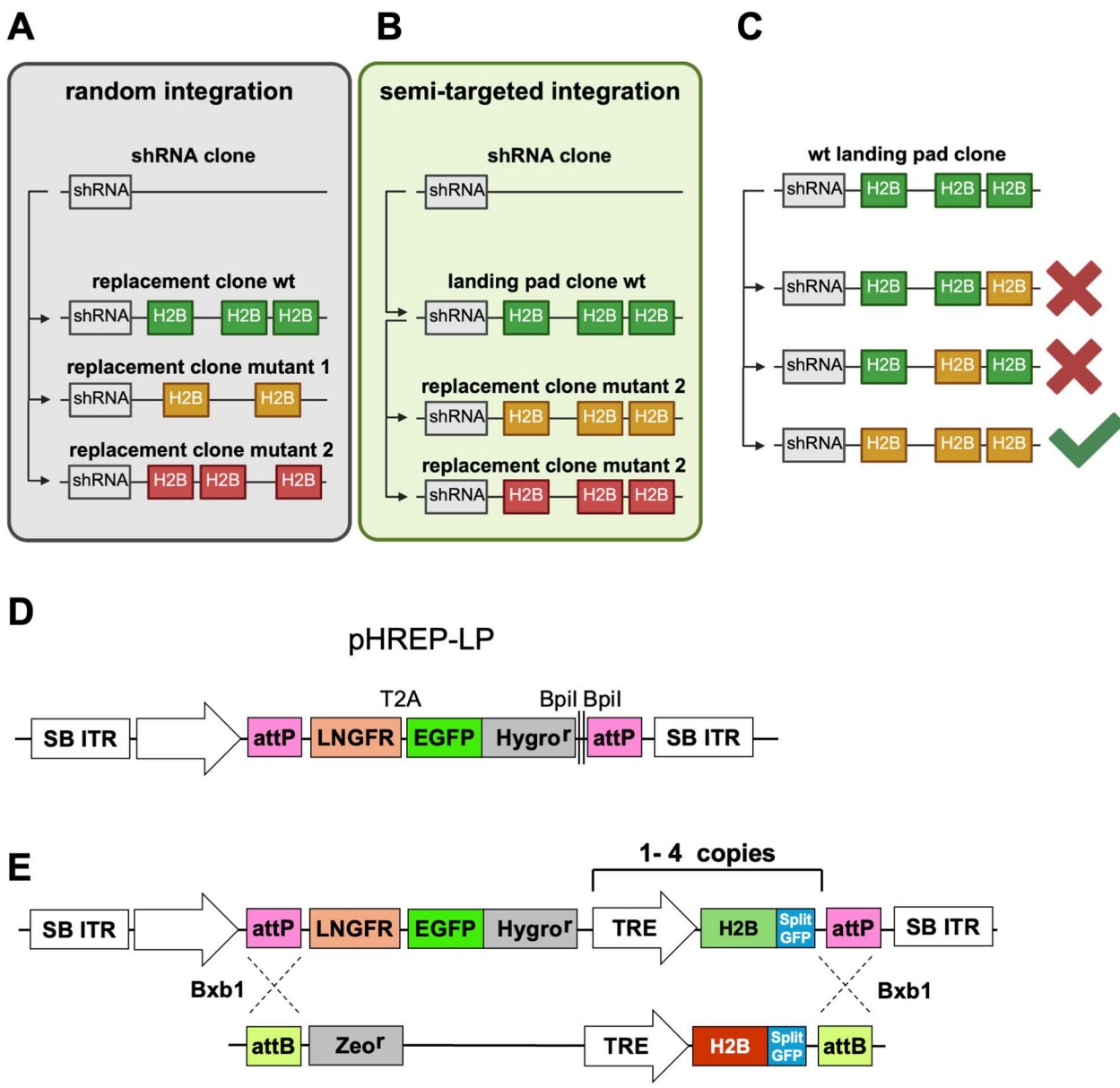

**Fig 6. Strategy for second-generation histone replacement by semi-targeted integration of replacement histone.** Schematic representation of the creation of replacement histone cell clones using different methods. **(A)** Random Integration: A parental shRNA knockdown cell clone is used to randomly integrate wild-type (wt) or mutant replacement histones. The integration copy number and locations vary between cell clones, reducing comparability between clones. **(B)** Semi-targeted Integration: A parental shRNA knockdown cell clone is used to randomly integrate wild-type replacement histone LPs. These LPs are subsequently targeted to replace the integrated wild-type histone with mutants at the same locus. This results in cell clones with identical locations and integration copy numbers, allowing full comparability between clones. **(C)** After RMCE-mediated cassette exchange, not all wild-type histone loci will be replaced by the mutant histones, necessitating negative selection against cells with incomplete exchange. **(D)** The second-generation pHREP-LP plasmid contains a constitutive promoter driving the expression of a truncated LNGFR surface protein, allowing sorting and enrichment of cells expressing this construct. A T2A site enables separate expression of a fusion protein between EGFP and the Hygromycin resistance gene. A cloning site allows integration of multiple copies of the shRNA-resistant replacement histone under the control of a tet-responsive promoter. **(E)** The pHREP-LP plasmid contains attP wild-type and attP-GA sites mediating enabling Bxb1-mediated RMCE [62,63]. This allows the replacement of the wild-type protein with any mutant carrying attB sequences on the coding plasmid. Cells with fully exchanged histones can be sorted by the loss of EGFP expression, enriched by the loss of LNGFR expression using magnetic beads, and gain a new selection marker through the insertion of the Zeocin resistance gene (Zeo^r).

**A**

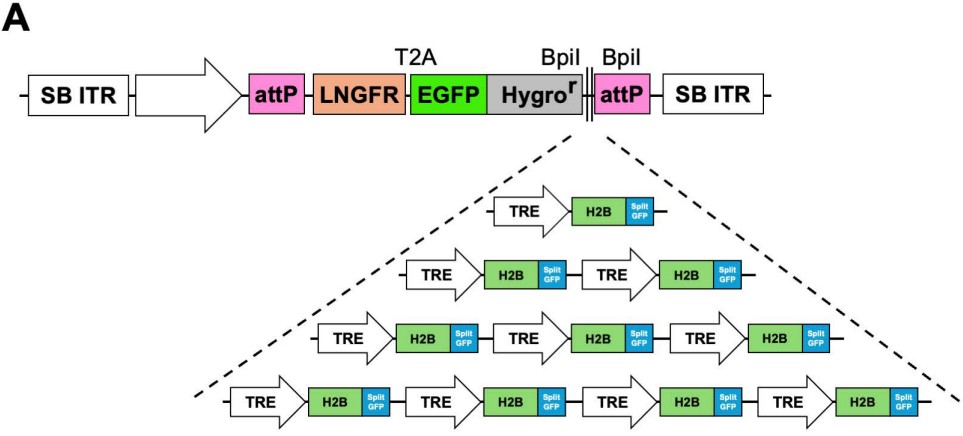

**B** **C**

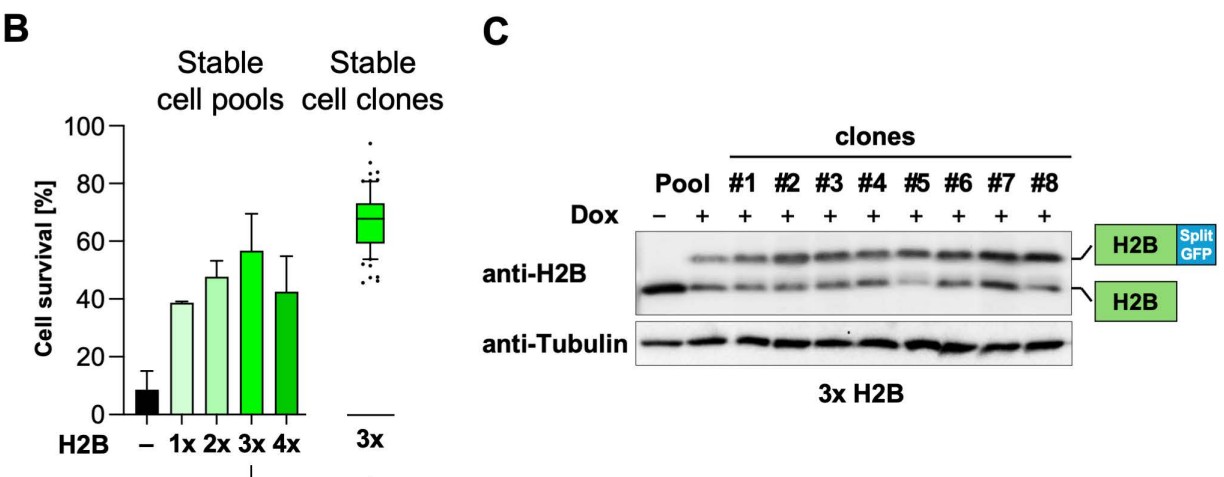

**D**

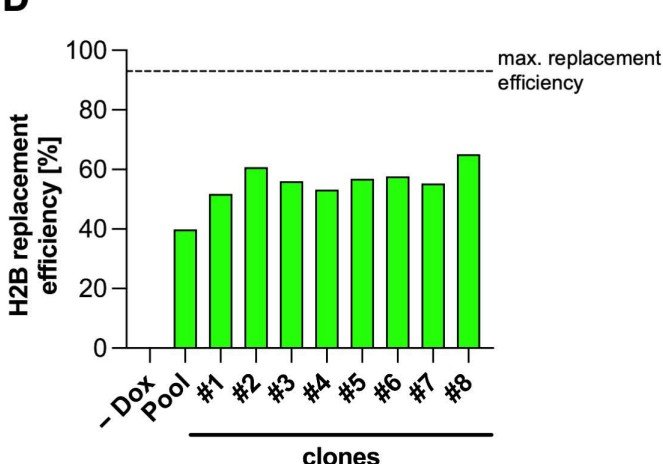

**Fig 7. Generation and analysis of stable inducible chicken H2B knockdown and re-expression clones in the second-generation system. (A)** Different copy numbers of TRE-driven cDNAs encoding H2B-splitGFP fusion proteins were inserted into the BpiI sites of pHREP-LP plasmid using Golden Gate assembly. **(B)** Cells suitable for shRNA knockdown (clone #2) were transfected with the different pHREP-LP plasmids shown in **(A)**,

containing 1 to 4 copies of shRNA-resistant wild-type replacement H2B. Subsequently, plasmid integration was selected with Hygromycin for 14 days. The resulting stable cell pools were treated with Dox for 4 days, and cell survival was measured by crystal violet staining and normalized to untreated controls. Data shown are from 3 independent biological replicates and are presented as mean±SD. The cell pool with 3 integrated H2B copies was used to produce single cell clones which were further selected with Dox and Puro for 4 days. The survival of the single cell clones was determined by crystal violet staining and normalized to uninduced controls. Data from 3 independent biological replicates are shown. Boxes indicate interquartile range (IQR), whiskers indicate 1.5 IQR and means are indicated by horizontal lines. **(C)** Eight selected stable H2B replacement clones and the parent pool were induced with Dox for 4 days or left untreated. Cell lysates were analyzed by Western blotting for the expression of endogenous and replacement H2B, a representative blot is shown. **(D)** The replacement efficiency of the blot shown in (C) was quantified as the ratio of the upper (replacement) histone H2B to the total H2B amount (replacement+endogenous H2B) and plotted.

from this cell pool by limited dilution. Single cell clones were tested for proliferation and survival in the presence of Dox. At the single-clone level, improved survival close to that of the uninduced cells was observed. Western blot experiments with selected clones consistently showed a significant knockdown of endogenous H2B and robust expression of the tagged H2B, although not reaching the level of endogenous H2B (Fig 7C). The replacement efficiency was lower than for the first-generation histone replacement system (Fig 7D).

Together these data show that the concept of a Dox-controlled histone replacement system in chicken cells is functional. Proof-of-principle experiments demonstrate that all canonical histones can be efficiently downregulated by a single shRNA each and that simultaneous re-expression of exogenously expressed histones restores cell survival. Further fine-tuning of various experimental parameters is required to isolate cell clones that achieve expression levels fully comparable to endogenous histones.

## Discussion

This work provides a proof-of-concept for a knockdown and re-expression approach for the replacement of histones in vertebrate cells. Here, we demonstrate that a single shRNA can efficiently knockdown the highly expressed histone proteins and show that integration of a few copies of an shRNA resistant histone can rescue this knockdown, as shown for H2B. This work also demonstrates that efficient replacement of endogenous histone can take place within a few days and that the replacement efficiency can reach almost 100%. We therefore propose that the exchange system presented here could be used as an additional tool for chromatin research. We believe that a comprehensive histone replacement system can augment current research on histone PTMs by providing the ability to directly interrogate the function of single amino acids or their modifications and even larger sequences.

Although chromatin research largely focuses on human and mouse models, histones are generally highly conserved in vertebrates and many modification sites of human histones are also found in chicken histones, enabling functional studies of these conserved amino acids. It may also be possible to functionally study human histones in their entirety in chicken cells, as human nucleosomes can function even in a small number of reconstituted yeast cells [54]. In principle the knockdown and re-expression approach shown here should also work in mouse or human cells when designing appropriate shRNAs and possibly combining multiple shRNAs to target all histone gene copies. Chicken fibroblasts, such as the DF-1 fibroblasts used here, are easy to culture and are highly transfectable. A particular feature of chicken cell culture is their growth at 39°C. The occasionally observed genetic instability of chicken fibroblasts in the presence of serum does not occur during long-term cultivation of the cells in appropriate culture media with limited serum concentrations [55].

When compared to the aforementioned replacement system relying on *in situ* mutation of histone genes [29], which retains all natural regulatory elements and the natural sequence context of the mutated genes, our concept relies fully on synthetic promoters and transcripts. An advantage of the concept presented here over the *in situ* mutation system is the experimental simplicity and time savings. However, using a "one shRNA fits all" approach and re-expression of a single replacement histone isoform also removes isoform diversity. While in chicken cells histone isoforms are very similar to

each other on a protein sequence level, the function of single amino acid differences are often still elusive. It is clear, however, that even changes of a few amino acids can confer specific functions as exemplified by the histone variant H3.3 [56]. Therefore, the system presented here could also be used to re-express different histone isoforms.

*In situ* editing is undeniably as close to physiological conditions as possible for a histone replacement system. This significant advantage however also comes with two drawbacks that depending on the application might need careful consideration. Firstly, CRISPR targeting of many genes is extremely labor intensive and therefore the approach suggested here is timesaving. Secondly, *in situ* mutation results in constitutive expression of the mutant histone, which – depending on the nature of the mutation – may be incompatible with cell survival and therefore unsustainable. Furthermore, the insertion of mutations by prime editing is limited to short editing lengths not exceeding 40 bp [57].

However, a few aspects need amendment and consideration to make the histone replacement system presented here fully functional. Firstly, the expression of endogenous histones occurs in the S-phase and is regulated at the RNA level by a histone stem loop (HSL) [58,59]. The addition of such an HSL to the re-expression histone cassette could enable S-phase-specific expression while still allowing inducibility by Dox. However, exchange of the polyA sequence by HSLs did not result in improved expression of the replacement histones (data not shown). Ideally, TetO Sequences would be positioned around or downstream of the TATA box of endogenous histone promoters to inhibit initiation and/or progression of transcription in the event of Tet repressor binding. The development of the tightly regulated Tet-On system, however, is promoter-specific and in the past required more than a decade of optimization [32], making this option beyond the scope of the present study. Additional optimization is warranted for the selection of the epitope tag used in the replacement histone. While useful in this context to distinguish endogenous from exogenous protein, a systematic and comparative investigation is required to determine which epitope tags are suitable for fusion with histones. Also, the relative amount of re-expressed histones should ideally be close to the expression level of the endogenous histone. In this context, cell survival alone will not be a sufficient criterion, as cells have been shown to survive for longer time with reduced histone content [60]. For this, careful measurements of histone levels need to be taken, for example by the inclusion of a sortable marker which allows for isolation of cells with appropriate expression levels. Another limitation of this (and all other) histone replacement systems is the potential compensation of the downregulated histone by non-canonical histone variants. However, we expect this effect to be minimal given the mutant histone is sufficiently re-expressed.

The next step would be to establish a suitable second-generation cell system with integrated wild-type versions of the different canonical histones. These cell systems will require comprehensive and deep phenotyping, including mapping the integration sites of the landing pads and characterizing the integration regions as safe harbor loci. Based on these cells, any interested laboratory could then introduce specific histone variants via RMCE and perform functional analyses to deepen our understanding of histone modifications.

## Materials and methods

### Cell culture and transfection

Cells were cultured in appropriate cell culture medium (DF-1 chicken fibroblasts: DMEM/F-12; HEK293T: DMEM) supplemented with 10% (v/v) FCS and Penicillin/Streptomycin in a humidified atmosphere with 5% (v/v) $CO_2$. DF-1 cells (ATCC CRL-12203) were grown at 39°C and HEK293T (ATCC CRL-3216) at 37°C. Cells were regularly passaged to maintain them in the logarithmic growth phase and checked for mycoplasma contamination by DNA staining and fluorescence microscopy. For cell seeding, cells were counted using a Logos LUNA-II automated cell counter with trypan blue for dead cell exclusion. Cells were seeded the day before transfection to reach 50% confluency at the time of transfection. The culture medium was replaced with medium containing 10% (v/v) FCS lacking antibiotics. HEK293T cells were transfected with PEI, and DF1 cells were transfected with Lipofectamine 3000. Stable cell lines were generated by adding appropriate antibiotics (Blasticidin 10 μg/mL, G418 400 μg/mL, Hygromycin B 50 μg/mL, Puromycin 1 μg/mL) one day post transfection to the medium.

## Program for shRNA scoring

The tool created for shRNA design is available via GitHub (github.com/ag-schmitz-gi/shRNA-designer;https://doi.org/10.5281/zenodo.18470380).

| Target | Sequence (5' to 3') |
|--------|---------------------|
| H1 | TTGGGCTTCACCGCCTTTGCCT |
| H2A | ATGCGCGTCTTCTTGTTGTCGC |
| H2B | TTGACGAACGAGTTCATGATGC |
| H3 | TTGGTGTCCTCGAAGAGCCCCA |
| H4 | TAGACCACGTCCATGGCCGTGA |

## Plasmid information

| Plasmid | Reference | Addgene # |
|---------|-----------|-----------|
| pHREP-KD-1 | This study | 240416 |
| pHREP-KD-1-shH2B | | 240417 |
| pHREP-RE-1-H2B | | 240419 |
| pHREP-KD-2 | | 240441 |
| pHREP-KD-2-shCtrl | | 240580 |
| pHREP-KD-2-shH1 | | 241994 |
| pHREP-KD-2-shH2A | | 241997 |
| pHREP-KD-2-shH2B | | 241996 |
| pHREP-KD-2-shH3 | | 241995 |
| pHREP-KD-2-shH4 | | 240431 |
| pHREP-LP | | 240432 |
| pHREP-LP-1xH2B | | 240433 |
| pHREP-LP-2xH2B | | 240438 |
| pHREP-LP-3xH2B | | 240439 |
| pHREP-LP-4xH2B | | 240440 |
| pSBtet-Neo | [40] | 60509 |
| pSBtet-Hyg | [40] | 60508 |
| pEGFP-N1 | Clontech | n/a |
| pMACS-LNGFR | Miltenyi Biotec | n/a |

## Generation of stable cell clones and determination of cell survival

The transfected cells were selected for several days to enrich stably transfected cells. In some cases, the cells were additionally sorted by FACS to further increase the proportion of stable transfectants. The selected cell pools were then seeded in a large dish at high dilution for clonal expansion of single cells or distributed into 96-well plates by limiting dilution. Clones were cultured until the colonies were large enough for picking or further analysis. For picking, the cells were briefly trypsinized for 60 seconds and quickly diluted with cold PBS. Colonies were then aspirated with a micropipette and transferred to a 96-well plate.

For qualitative determination of cell survival, the medium was aspirated, and the wells were washed twice with ice-cold PBS and placed on ice. Cells were fixed with ice-cold methanol for 10 minutes at room temperature. Subsequently, the cells were stained with staining solution (0.5% (w/v) crystal violet, 25% (v/v) methanol, $H_2O$) for 10 minutes at room temperature. After staining, the cells were rinsed with tap water until no more crystal violet was released. The blue staining of the air-dried wells was scanned. For the quantitative determination of cell survival, the cells were stained and washed as described above, followed by a 30-minute de-staining in a 2% (w/v) SDS solution. The absorbance at 600 nm was then measured using a GloMax plate reader.

## Statistical analysis

The number of biological replicates is indicated in the respective figure legends. Means are shown with standard deviations, represented by error bars, data were analyzed using GraphPad Prism 9.

## Western blotting and RT-qPCR

The medium was removed, and the cells were washed with cold PBS. The cells were scraped off in a small volume of cold PBS and collected. The culture vessels were rinsed again with cold PBS, and the cells were pelleted by centrifugation at 500 rcf for 5 minutes at 4 °C. The cell pellet was lysed in five times its volume of 1x SDS sample buffer. The lysates were then sonicated three times for 20 seconds in continuous mode to shear the DNA and fully solubilize the cells. The samples were boiled for 5 minutes at 95 °C, and the proteins were separated by SDS-PAGE and analyzed by Western blotting. For immunodetection, proteins were transferred from the SDS gels to polyvinylidene fluoride (PVDF) membranes using a semi-dry blotting device as described [61].

To determine relative mRNA expression by RT-qPCR, RNA was first isolated using the NucleoSpin° Kit (Macherey-Nagel, cat. no. 740955). The RNA was eluted in 30–50 μl RNase-free $H_2O$, and the RNA concentration was measured by determining the optical density at 260 nm ($OD_{260}$) using an Eppendorf photometer and a microliter cuvette. To generate cDNA, the PrimeScript RT Reagent Kit (Takara, cat. no. RR037B) was used. The cDNA was diluted and employed for quantitative PCR reactions using SYBR Green as a reporter on a StepOnePlus Real-Time PCR System (Applied Biosystems). A melting curve analysis was performed for each primer pair to confirm reaction specificity.

## Supporting information

**S1 Fig. Testing of shRNAs against all canonical chicken histones.** Expression vectors for EGFP fused with the respective canonical chicken histones were generated. The individual fusion proteins were co-expressed in HEK293T cells with plasmids encoding different histone shRNAs or a control shRNA. After 2 days, total RNA was extracted, and mRNA levels were measured by qPCR against EGFP and normalized to the control. Data from 3 independent biological replicates are shown as means ± SD, with all samples normalized to their respective shRNA control. The shRNAs highlighted in red were used for further experiments.
(PDF)

**S1 File. Western Blot image Raw Data.**
(PDF)

**S1 Table. shRNAs tested targeting chicken histone genes.**
(XLSX)

## Acknowledgments

We acknowledge expert technical assistance by Yvonne Horn and Markus Schwinn.

## Author contributions

**Conceptualization:** Maximilian Pfisterer, M. Lienhard Schmitz.

**Data curation:** M. Lienhard Schmitz.

**Formal analysis:** M. Lienhard Schmitz.

**Funding acquisition:** M. Lienhard Schmitz.

**Investigation:** Maximilian Pfisterer, Amelie Pritz, Anna Parry.

**Methodology:** Maximilian Pfisterer.

**Project administration:** M. Lienhard Schmitz.

**Supervision:** M. Lienhard Schmitz.

**Visualization:** Maximilian Pfisterer.

**Writing – original draft:** M. Lienhard Schmitz.

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
