## [Decision Letter · Decision Letter 0]

15 Dec 2025

Dear Dr. Schmitz,

Thank you for submitting your manuscript to PLOS ONE. After careful consideration, we feel that it has merit but does not fully meet PLOS ONE’s publication criteria as it currently stands. Therefore, we invite you to submit a revised version of the manuscript that addresses the points raised during the review process.

We look forward to receiving your revised manuscript.

Kind regards,

Zu Ye, Ph.D.

Academic Editor

PLOS One

Journal Requirements:

DFG grant 416910386

4. Please expand the acronym “DFG” (as indicated in your financial disclosure) so that it states the name of your funders in full.

5. Please provide a complete Data Availability Statement in the submission form, ensuring you include all necessary access information or a reason for why you are unable to make your data freely accessible. If your research concerns only data provided within your submission, please write "All data are in the manuscript and/or supporting information files" as your Data Availability Statement.

Reviewers' comments:

Reviewer's Responses to Questions

**Comments to the Author**

1. Is the manuscript technically sound, and do the data support the conclusions?

Reviewer #1: Yes

Reviewer #2: Yes

Reviewer #3: Yes

2. Has the statistical analysis been performed appropriately and rigorously?

Reviewer #1: Yes

Reviewer #2: Yes

Reviewer #3: Yes

3. Have the authors made all data underlying the findings in their manuscript fully available?

Reviewer #1: Yes

Reviewer #2: Yes

Reviewer #3: Yes

4. Is the manuscript presented in an intelligible fashion and written in standard English?

Reviewer #1: Yes

Reviewer #2: Yes

Reviewer #3: Yes

Reviewer #1: I am enthusiastic about this paper. The goal is to develop a system such that histones can be replaced by chosen mutants. This is important for research, but quite challenging to do in metazoans because the genes are present in many copies. Here, the authors settle on a chicken system, because it has fewer genes, and they are more similar to one another. They then combine inducible shRNA and inducible re-expression system by using clever molecular biology approaches. The paper is very clearly written. The data are clear, correctly controlled, and interpreted. I cannot think of a major correction to make to this paper.

Reviewer #2: The manuscript “A novel approach towards a histone replacement system in Tetrapods” by Pfisterer et al describes a method to replace canonical histones in chicken DF-1 cells. Authors design and test several constructs in order to express shRNAs specific for all canonical histones in chicken and show that even a single shRNA efficiently deplete their expression as shown in WB analyses. Subsequently, they design and test some other constructs that re-express the KD histone in a form resistant to shRNA under Dox induction in a tagged form to assess for replacement.

Overall, and as a proof of principle, the paper is sound, results are well presented and the caveats and possibilities to improve the system accurately discussed.

While the manuscript is fine and should be acceptable for publication in PLOS ONE I have some points that authors should clarify:

I think that instead of using an inducible promoter to restore the depleted histones that, as authors indicate, does not respect the S-phase natural expression of histones, it would be much more accurate to express them from their natural promoters in a multicopy construct. In this way any potentially perturbing effects of expression out of S-phase would be bypassed at the same time that expression will likely result in the correct amount of protein expressed. Have the authors tried this approach?

In agreement with the authors, I think it is a good approach to introduce landing pads to minimize random integration side effects. However, to make them trustable the landing pads positions in the genome should be mapped and those in the best positions (those having minimal effects) selected.

On one hand, authors indicate that the copy number (but also the variants) of the different histones may be a problem in vertebrates other than chicken. However, when they tested it in human HEK293T cells the approach seemed to work properly, as shown at the mRNA level. It would be very informative if semiquantitative WB analyses for these experiments are included.

Minor points:

On Table 1 there is a mistake. Human histone H1 are 7 somatic isoforms (+4 germline specific isoforms)

In Drosophila the histone replacement systems designed by the group of Robert Duronio (McKay et al, (2015) Dev Cell32, 373-386 and a CRISPR-CAS9 based approach by the group of Guanjun Gao (Zhang, W. et al. (2019) Dev. Cell 48, 406–419) should be also referenced in the introduction.

There are a few typing mistakes

Reviewer #3: This manuscript by Pfisterer et al. describes a method for partial histone replacement in a tetrapod system. Histone replacement has been used to great effect in yeast and Drosophila to test the functional role of histone modifications. In these model systems, histones genes are conveniently encoded together in a single genomic locus, permitting complete knockout of endogenous histones. These knockouts can be rescued through exogenous expression of either wild type or mutant histones, enabling functional comparisons. In higher eukaryotes complete histone replacement is challenging because histones are encoded by many genes in clusters that are dispersed through the genome. This complicates targeting methods for gene knockout and necessitates an alternative strategy. To this end, the authors introduce an shRNA-mediated partial replacement system. The first generation utilizes an optimized, inducible shRNA to knockdown endogenous histone, while the replacement histone is expressed from a sleeping beauty transposon system. As proof of principle, the authors target canonical histone H2B in chicken fibroblasts, given the relatively few H2B genes in chicken and the high sequence similarity between them, which facilitates shRNA design. Based on western blot analysis, the authors report efficient suppression of endogenous H2B, with corresponding expression of replacement H2B that approximates endogenous levels. A potential issue with the first-generation system is that comparison between cell lines with different replacement histone is challenging (e.g., comparison between replacement with wild type histone H2B and a mutant version) because it is not possible to control the number or location of transposon insertions. The authors therefore develop a second-generation system that uses landing pads to standardize integration of the replacement system. Though the second-generation system overcomes some limitations in the first-generation system, it is far less efficient, as the ratio of exogenous to endogenous histone appears much lower. Overall, the manuscript is well written and clear. A substantial portion of the manuscript is dedicated to describing the system, with modest experimental data. The approach is elegant, though the applications are limited given its dependence on shRNA. Many of these limitations are discussed in the text, which is important. Detailed comments are below.

-The authors choose to target canonical histones, which is reasonable; however, variant histones are only mentioned in passing and not discussed in the context of the replacement system. The non-canonical histone variants could be an important consideration for many of the histones, as they will likely not be targeted by the shRNAs given sequence differences between the genes. It is possible that the non-canonical histones could compensate for endogenous histones targeted by the shRNA knockdown and complicate interpretation of the resulting data. This would be an important point to highlight in the manuscript.

-The authors demonstrate downregulation of endogenous histone H2B by western blot, which is efficient but incomplete in the first-generation system and relatively inefficient in the second-generation system. The western analyses do not provide information on the genomic localization of remaining endogenous histone and it is possible that the endogenous histone is not uniformly distributed across the genome, but rather collects at higher levels in certain regions compared to others (perhaps even at wild type levels). This could be important for comparative analyses since one could imagine that certain genes may retain endogenous wild type histone rather than a mutant replacement histone, which would have the potential to skew both gene expression and resulting phenotypes. Ideally, the authors would show genome-wide distribution of the endogenous histone in the replacement system; however, this would be challenging since antibodies against H2B would not distinguish between endogenous and exogenous histone. The authors could infer levels of endogenous histone by mapping total histone and exogenous histone based on the epitope tag. At a minimum, this complication should be discussed in the text.

-The authors use 96-well plates to screen for clones that survive and proliferate following histone replacement. On page 10, the authors note that “most clones showed poor survival.” Is there any concern that the authors are selecting for clones with an abnormal karyotype that bestows a growth advantage? The puro selection is an intentional effort to identify clones that highly express the replacement histone but the puro resistance gene should be effective at very low levels. The persistence of only four clones seems to indicate a harsh selective process from initial histone knockdown. Ensuring that the cells are not abnormal as a result of this process could be important for studies attempting to associate histone modifications with cell physiology.

-As noted above, knockdown of endogenous histone is inefficient in the second generation system. The authors should provide replacement efficiencies as shown in Figure 4C and discuss the implications if replacement is weak.

Minor point

There are different kinds of histone replacement systems (i.e., partial vs complete), which are discussed in PMID: 35853806. This is a minor semantics point, but it seems that the system outlined in this manuscript should be defined as partial to avoid confusion with complete systems.

**Do you want your identity to be public for this peer review?** For information about this choice, including consent withdrawal, please see our Privacy Policy

Reviewer #1: No

Reviewer #2: No

Reviewer #3: No

---

## [Author Response · Author response to Decision Letter 1]

13 Jan 2026

Re. Revision of PONE-D-25-37538

Dear Dr. Ye,

Thank you very much for sending us the referees` comments on our manuscript by Pfisterer et al. entitled "A novel approach towards a histone replacement system in Tetrapods". In the meantime we have revised the manuscript according to the suggestions of the reviewers. In the following section, we will provide a point-to-point answer to the questions, criticism and queries raised by the two referees, who really helped to amend this manuscript. Our responses to the reviewers written in blue:

Reviewer #1:"I am enthusiastic about this paper..."

Answer: Thank you very much for these positive comments.

Reviewer #2:

1) I think that instead of using an inducible promoter to restore the depleted histones that, as authors indicate, does not respect the S-phase natural expression of histones, it would be much more accurate to express them from their natural promoters in a multicopy construct. In this way any potentially perturbing effects of expression out of S-phase would be bypassed at the same time that expression will likely result in the correct amount of protein expressed. Have the authors tried this approach?

Answer: We fully agree with the concern that S-phase-specific expression of the replacement histone would more faithfully recapitulate the endogenous histone expression. We therefore inserted histone stem loops, but this did not improve expression. A second option would be the insertion of Tet operators upstream of the cell cycle-specific histone promoters. Historically, however, the development of the Tet-On system and the optimization of all the different parameters ensuring the tightness and strong inducibility of the system took more than a decade (Paillard et al. Hum Gene Ther. 1998, PMID: 9607408; Colicchia et al. FEBS Open Bio. 2022, PMID: 36062323) and thus lies beyond the scope of this study. We have addressed all these aspects in the revised discussion.

2) In agreement with the authors, I think it is a good approach to introduce landing pads to minimize random integration side effects. However, to make them trustable the landing pads positions in the genome should be mapped and those in the best positions (those having minimal effects) selected.

Answer: We fully agree with this statement. However, since this is merely a proof-of-principle study, we have only determined the copy number of integration sites, but not their genomic locations. A complete characterization would also require to asses whether the integrations are bona fide safe harbor sites: genetically stable locations that are not subject to potential (epi)genetic changes and not showing interactions with other genomic loci. In accordance with the reviewer’s suggestions, these aspects have been included in the revised discussion.

3) On one hand, authors indicate that the copy number (but also the variants) of the different histones may be a problem in vertebrates other than chicken. However, when they tested it in human HEK293T cells the approach seemed to work properly, as shown at the mRNA level. It would be very informative if semiquantitative WB analyses for these experiments are included.

Answer: We would like to clarify this point in more detail: In the experiment, the endogenous histones of 293T cells were not downregulated. As described in the manuscript, EGFP-tagged chicken histones were coexpressed with the different shRNAs against the chicken histones in 293T cells to provide a rapid and quantifiable system for the characterization of all the various shRNAs.

Minor points:

On Table 1 there is a mistake. Human histone H1 are 7 somatic isoforms (+4 germline specific isoforms)

Answer: Thank you for spotting this mistake, the table has been corrected accordingly.

In Drosophila the histone replacement systems designed by the group of Robert Duronio (McKay et al, (2015) Dev Cell32, 373-386 and a CRISPR-CAS9 based approach by the group of Guanjun Gao (Zhang, W. et al. (2019) Dev. Cell 48, 406–419) should be also referenced in the introduction.

Answer: The reference from McKay and coworkers was added, the paper by Zhang et al. was already cited.

There are a few typing mistakes.

Answer: We have carefully read through the text again and tried to identify and correct all errors.

Reviewer #3:

1) The authors choose to target canonical histones, which is reasonable; however, variant histones are only mentioned in passing and not discussed in the context of the replacement system. The non-canonical histone variants could be an important consideration for many of the histones, as they will likely not be targeted by the shRNAs given sequence differences between the genes. It is possible that the non-canonical histones could compensate for endogenous histones targeted by the shRNA knockdown and complicate interpretation of the resulting data. This would be an important point to highlight in the manuscript.

Answer: We have added these considerations, which apply to all histone replacement systems, to the discussion.

2) The authors demonstrate downregulation of endogenous histone H2B by western blot, which is efficient but incomplete in the first-generation system and relatively inefficient in the second-generation system. The western analyses do not provide information on the genomic localization of remaining endogenous histone and it is possible that the endogenous histone is not uniformly distributed across the genome, but rather collects at higher levels in certain regions compared to others (perhaps even at wild type levels). This could be important for comparative analyses since one could imagine that certain genes may retain endogenous wild type histone rather than a mutant replacement histone, which would have the potential to skew both gene expression and resulting phenotypes. Ideally, the authors would show genome-wide distribution of the endogenous histone in the replacement system; however, this would be challenging since antibodies against H2B would not distinguish between endogenous and exogenous histone. The authors could infer levels of endogenous histone by mapping total histone and exogenous histone based on the epitope tag. At a minimum, this complication should be discussed in the text.

Answer: This point is well taken and overlaps with the comment from Reviewer #3, point 2. Accordingly, we have addressed these important points in the revised manuscript. The revised version considers two distinct levels:

a) At the genomic level, these cell systems will require comprehensive and deep phenotyping, including mapping the integration sites of the landing pads and characterizing the integration regions as safe harbor loci: genetically stable locations that are not subject to potential (epi)genetic changes and do not interact with other genomic loci.

b) At the histone level, it is important to distinguish between endogenously and exogenously expressed histones. In our system, this was achieved through the addition of epitope tags, which allow discrimination between endogenous and re-expressed histones at both the biochemical and genetic levels.

We suggest that such experiments are best suited for follow-up studies building on this proof-of-principle work.

3) The authors use 96-well plates to screen for clones that survive and proliferate following histone replacement. On page 10, the authors note that “most clones showed poor survival.” Is there any concern that the authors are selecting for clones with an abnormal karyotype that bestows a growth advantage? The puro selection is an intentional effort to identify clones that highly express the replacement histone but the puro resistance gene should be effective at very low levels. The persistence of only four clones seems to indicate a harsh selective process from initial histone knockdown. Ensuring that the cells are not abnormal as a result of this process could be important for studies attempting to associate histone modifications with cell physiology.

Answer: We have revised and improved our text, which was indeed somewhat ambiguously phrased, so that this aspect is now more clearly conveyed. It is now described that stable integration of the plasmids for shRNA knockdown is achieved via the neomycin resistance gene, while the addition of puromycin selects for tetracycline-inducible shRNA expression. This approach led to the isolation of four cell clones that ceased to proliferate upon addition of Dox, whereas they grew normally and comparably to wild-type cells in the absence of Dox. The relatively small number of obtained cell clones is likely due to the requirement for very high levels of inducible shRNA expression to downregulate the endogenous histone. As reported in the literature, a bona fide proliferation defect occurs only when large amounts of the histone are depleted (Corcoran et al. Plant Cell 2022, PMID: 35879829)

4) As noted above, knockdown of endogenous histone is inefficient in the second generation system. The authors should provide replacement efficiencies as shown in Figure 4C and discuss the implications if replacement is weak.

Answer: As suggested by the reviewer, we quantified the H2B replacement efficiency, which is now shown in the new Fig. 7D. The results are discussed in the revised version of the manuscript.

Minor point

There are different kinds of histone replacement systems (i.e., partial vs complete), which are discussed in PMID: 35853806. This is a minor semantics point, but it seems that the system outlined in this manuscript should be defined as partial to avoid confusion with complete systems.

Answer: This point is correct, and we therefore now state that a partial histone replacement system has been developed.

We thank the reviewers for the time and effort they have invested to further improve the conceptual, technical and didactical parts of the manuscript. All authors have seen the final version of the manuscript and concur with its content We are confident that we have addressed all major issues and hope that the manuscript has experienced strong amendment so that it can be considered further for publication.

With best wishes,

Lienhard Schmitz & Maximilian Pfisterer

---

## [Editor Report · Decision Letter 1]

15 Jan 2026

A novel approach towards a histone replacement system in Tetrapods

PONE-D-25-37538R1

Dear Dr. Schmitz,

We’re pleased to inform you that your manuscript has been judged scientifically suitable for publication and will be formally accepted for publication once it meets all outstanding technical requirements.

Kind regards,

Zu Ye, Ph.D.

Academic Editor

PLOS One
---

## [Editor Report · Acceptance letter]

PONE-D-25-37538R1

PLOS One

Dear Dr. Schmitz,

I'm pleased to inform you that your manuscript has been deemed suitable for publication in PLOS One. Congratulations! Your manuscript is now being handed over to our production team.

Kind regards,

on behalf of

Prof. Zu Ye

Academic Editor

PLOS One